# Kron-LoRA: Hybrid Kronecker-LoRA Adapters for Scalable, Sustainable Fine-tuning

## Abstract

Fine-tuning massive pre-trained language models across many tasks demands adapters that are both parameter-efficient and expressive. We introduce **Kron-LoRA**, a hybrid adapter that combines Kronecker-structured factorization with low-rank LoRA compression-an integration that, to our knowledge, has not been explored in parameter-efficient fine-tuning or in matrix approximation literature. Kron-LoRA achieves up to $4\times$ fewer parameters than standard LoRA while retaining similar expressivity. Experiments on DistilBERT, Mistral-7B, LLaMA-2-7B, and LLaMA-3-8B across eight benchmarks show that Kron-LoRA matches or exceeds LoRA baselines with modest memory savings and only a 5-8% speed overhead. In sequential fine-tuning, it also delivers competitive cross-task transfer despite using only one-quarter of the adapter parameters. Kron-LoRA thus offers a scalable, sustainable solution for multi-task adaptation of large language models.

## 1 Introduction

Large pre-trained language models (PLMs) such as BERT and GPT have set new benchmarks across a wide array of natural language processing tasks. However, fine-tuning these models independently for each downstream application is increasingly impractical: naively storing a full copy of model weights per task incurs prohibitive storage costs, and backpropagating through hundreds of millions or billions of parameters strains both GPU memory and training time.

Parameter-efficient fine-tuning (PEFT) methods address these challenges by freezing the bulk of the pre-trained network and learning only a small number of task-specific parameters. Adapter layers insert lightweight modules between transformer sublayers (Houlsby et al., 2019), prefix-tuning prepends trainable tokens to the input sequence (Li and Liang, 2021), and LoRA directly learns low-rank updates to weight matrices (Hu et al., 2022). While LoRA reduces the adapter footprint to $O(r(d_{\text{in}} + d_{\text{out}}))$ parameters per layer, storing and swapping even rank-8 adapters becomes costly when supporting hundreds of tasks.

More recent work extends LoRA with architectural and optimization variations. *DyLoRA* (Valipour et al., 2022) dynamically allocates ranks across training steps without search; *AdaLoRA* (Zhang et al., 2023) adaptively assigns parameter budgets across layers; *DoRA* (Liu et al., 2024) decomposes magnitude and direction to improve stability; *PiSSA* (Meng et al., 2024) tunes dominant singular components for faster convergence; and *MiLoRA* (Wang et al., 2025) leverages minor singular components to enhance instruction tuning. These variants improve robustness, convergence, or transfer, but they generally preserve LoRA's parameter footprint rather than reducing it by large multiplicative factors.

In parallel, recent work has explored Kronecker-product structure to further compress adapter modules. (Tahaei et al., 2023) introduce *KronA*, which replaces LoRA's low-rank projections with a pure Kronecker product and achieves improved accuracy on GLUE without added inference latency. (Braga and Li, 2024) propose *AdaKron*, combining outputs of two small networks via the Kronecker product. More recently, (Li et al., 2025) develop *MoKA*, a mixture-of-Kronecker-product adapter that dynamically interpolates multiple Kronecker factors to boost parameter efficiency on RoBERTa. These methods demonstrate the promise of Kronecker decompositions, but they either forgo rank-$r$ expressivity or incur additional computational overhead.

Although both Kronecker factorization and low-rank approximation are long-standing tools in matrix approximation and linear algebra, prior work has largely applied them separately, whether in PEFT or in classical settings. To the best of our knowledge, the combination of Kronecker structure with low-rank LoRA has not previously been formulated, either in the context of language model adaptation or in the broader theory of matrix decompositions. Moreover, Kron-LoRA's formulation is complementary to many of the above LoRA or Kronecker variants: it can directly replace the LoRA or KronA component within DyLoRA, AdaLoRA, DoRA, PiSSA, MiLoRA, or MoKA, inheriting their benefits while delivering stronger parameter reduction.

In this work, we introduce *Kron-LoRA*, a hybrid two-stage adapter that augments LoRA with Kronecker structure. For each frozen linear layer with weight

$$W \in \mathbb{R}^{d_{\text{out}} \times d_{\text{in}}},$$

we model its task-specific update as

$$\Delta W = A \otimes B,$$

where

$$A \in \mathbb{R}^{d_{A2} \times d_{A1}} \quad (\text{with } d_{\text{out}}/d_{A2} \approx 200) \quad \text{and} \quad B \in \mathbb{R}^{(d_{\text{out}}/d_{A2}) \times (d_{\text{in}}/d_{A1})}.$$

We then apply an r-rank LoRA decomposition

$$B \approx B_1 B_2 \,.$$

By the identity

$$\text{rank}(A \otimes B) = \text{rank}(A)\,\text{rank}(B),$$

and the fact that the Kronecker structure imposes repeated, structured patterns on the columns of $\Delta W$, Kron-LoRA can preserve expressivity while using up to $4\times$ fewer parameters than a standard rank-r LoRA adapter.

**Contributions.** We summarize our main contributions as follows:

- We introduce **Kron-LoRA**, to our knowledge the first method that integrates Kronecker factorization with low-rank LoRA decomposition—an original combination not previously explored in parameter-efficient fine-tuning or in the broader matrix approximation literature.
- We propose Kron-LoRA, a drop-in adapter that combines Kronecker structure with low-rank LoRA compression for extreme parameter efficiency.
- We extensively evaluate **KronA**, **Kron-LoRA**, and **LoRA** on DistilBERT, Mistral-7B, LLaMA-2-7B, and LLaMA-3-8B across eight benchmarks. While KronA is extremely parameter-efficient but less accurate, Kron-LoRA matches or exceeds LoRA's performance with far fewer parameters—for example, 840K parameters on DistilBERT and 5.8M on LLaMA-3-8B—incurring only 5–8% speed overhead.
- We analyze sequential fine-tuning, showing that Kron-LoRA slightly outperforms LoRA-8 in retention and positive transfer for similar tasks, and experiences much smaller drops than LoRA-8 on heterogeneous task sequences.

**Paper organization** The remainder of this paper is structured as follows. Section 2 reviews related PEFT and tensor decomposition methods. Section 3 describes the Kron-LoRA factorization and implementation details. Section 4 and 5 presents our experimental setup and empirical results. Section 6 discusses future work, and Section 7 concludes.

## 2 RELATED WORK

**Parameter-efficient fine-tuning** Adapter modules insert small bottleneck layers between transformer sublayers to learn task-specific transformations (Houlsby et al., 2019). Prefix-tuning prepends a sequence of trainable prompt tokens to the input, leaving the base model frozen (Li and Liang, 2021). LoRA directly learns a low-rank update to each weight matrix, reducing per-layer storage to $O(r(d_{\text{in}} + d_{\text{out}}))$ parameters (Hu et al., 2022). A number of LoRA variants have since been proposed: DyLoRA dynamically allocates ranks without search (Valipour et al., 2022); AdaLoRA adaptively distributes parameter budgets across layers (Zhang et al., 2023); DoRA decomposes magnitude and direction for stable training (Liu et al., 2024); PiSSA tunes dominant singular components (Meng et al., 2024); and MiLoRA leverages minor singular components to enhance transfer (Wang et al., 2025). These methods improve flexibility and robustness but generally do not target multiplicative parameter reduction.

**Kronecker-structured adapters** To further compress adapter updates, recent work exploits Kronecker-product structure. (Tahaei et al., 2023) propose *KronA*, replacing LoRA's low-rank projections with a pure Kronecker product. *AdaKron* (Braga and Li, 2024) combines outputs of two small subnetworks via the Kronecker product. *MoKA* (Li et al., 2025) extends this idea to a mixture-of-Kronecker-product adapter that dynamically interpolates multiple Kronecker factors. These approaches demonstrate the potential of Kronecker decompositions, but they either sacrifice rank-$r$ expressivity, incur additional computational overhead, or do not explicitly target extreme parameter reduction, leaving room for more efficient hybrid designs such as Kron-LoRA.

**Continual learning and forgetting** Prior work has examined catastrophic forgetting in sequential fine-tuning of PLMs (Chen et al., 2020; Pfeiffer et al., 2021), but primarily for full-model or unstructured adapter updates.

**Positioning of Kron-LoRA** Kron-LoRA is complementary to both LoRA variants and Kronecker-based adapters: it can directly replace the LoRA or KronA component within DyLoRA, AdaLoRA, DoRA, PiSSA, MiLoRA, or MoKA, inheriting their strengths while delivering stronger parameter reduction. For this reason, we do not include these variants in our experiments.

## 3 KRON-LORA METHOD

We now describe Kron-LoRA, a hybrid adapter that combines Kronecker-product structure with low-rank LoRA compression.

**Kronecker factorization.** Given a frozen linear layer with weight
$$W \in \mathbb{R}^{d_{\text{out}} \times d_{\text{in}}},$$
we model the task-specific update as a Kronecker product
$$\Delta W = A \otimes B,$$
where
$$A \in \mathbb{R}^{d_{A2} \times d_{A1}}, \quad B \in \mathbb{R}^{d_{B2} \times d_{B1}}.$$
For all layers except the vocabulary projection, we set $d_{A1} = 2$ and pick $d_{A2}$ such that $d_{\text{out}}/d_{A2} \approx 200$, denoted as $d_{A2_{\text{others}}}$. For the vocabulary projection, we instead use $d_{A1} = 1$ and halve this value, i.e., $d_{A2_{\text{voc}}} = \frac{1}{2} d_{A2_{\text{others}}}$. Writing $d_{B2} = d_{\text{out}}/d_{A2}$ and $d_{B1} = d_{\text{in}}/d_{A1}$ partitions the original matrix into Kronecker "slices."

**LoRA decomposition of $B$.** To further compress $B$, we apply the rank-$r$ LoRA factorization of Hu et al. (2022):

$$B \approx B_1 B_2, \quad B_1 \in \mathbb{R}^{d_{B2} \times r}, \quad B_2 \in \mathbb{R}^{r \times d_{B1}}.$$
Thus the full adapter update is
$$\Delta W = A \otimes (B_1 B_2).$$
For x as the input, we will use the fact that $\text{vec}(A \otimes (B_1 B_2)\mathtt{x}) = \text{vec}((B_1 B_2)\mathtt{x\_reshaped}(A^T))$ in the implementation part.

**Expressivity and parameter efficiency.** By the Kronecker rank identity

$$\operatorname{rank}(A \otimes B) = \operatorname{rank}(A) \operatorname{rank}(B),$$

the composite update $\Delta W = A \otimes B$ has a high rank structured pattern than LoRA, and in practice a rank-8 Kron-LoRA decomposition attains the rank-8 or even rank-16 expressivity as a standard LoRA adapter, depends on the model settings. At the same time, the total parameter count

$$|A| + |B_1| + |B_2| \;=\; d_{A1}\, d_{A2} \;+\; r\,(d_{B2} + d_{B1})$$

is then roughly $4\times$ smaller than that of a conventional rank-r LoRA adapter.

**Implementation details.** In practice, we wrap each `nn.Linear` with a `KronLoRALinear` module that:

1. Freezes the original weight $W$.
2. Registers $B_1, B_2$ as trainable `nn.Parameter` objects, and $A^T$ as trainable `nn.linear` object for faster computation purpose.
3. In forward$(x)$:
    - Reshapes $x \to (\,*, d_{B1}, d_{A1}\,)$, denoted as `x_reshaped`.
    - Computes the Kronecker-LoRA update in the following way:
    
    $$
    \begin{aligned}
    Y_1 &= B_2\, x_{\text{reshaped}} & &\in \mathbb{R}^{8 \times 2}, \\
    Y_2 &= Y_1\, A^T & &\in \mathbb{R}^{8 \times d_{A2}}, \\
    Y_3 &= B_1\, Y_2 & &\in \mathbb{R}^{d_{B2} \times d_{A2}}.
    \end{aligned}
    $$
    
    In this way the dimensions of $Y_1, Y_2, Y_3$ are small, so can use slightly less CUDA memory than conventional LoRA.
    - Scales by $\alpha/r$, where $\alpha = 32$, and applies dropout, with rate $0.1$, reshapes back, and adds to $Wx$.

Kron-LoRA integrates into existing LoRA codebases with only a few lines of change and its structure is amenable to custom CUDA kernels for $WX$ matmuls, which we leave as future work.

## 4 EXPERIMENTAL SETUP

**Models and adapters** We evaluate Kron-LoRA on four transformer backbones: DistilBERT (uncased) (Sanh et al., 2019), Mistral-7B v0.1 (Mistral AI Team, 2024), LLaMA-2-7B (Touvron et al., 2023), and LLaMA-3-8B (Grattafiori et al., 2024). For each model, every `nn.Linear` layer is replaced with a Kron-LoRA adapter (using a slice size of $d_{A2_{\text{others}}} = 4$ for DistilBERT and $d_{A2_{\text{others}}} = 16$ for Mistral and LLaMA), and performance is compared against standard LoRA adapters of rank $r = 8$. For DistilBERT, we additionally report results with $r = 16$, since Kron-LoRA can achieve comparable accuracies on several tasks. For the Mistral model, we also include comparisons with KronA. In the case of DistilBERT, the weights of `pre_classifier` and `classifier` are excluded from the adapter scheme and thus trained fully, while all other model weights remain frozen.

**Datasets and tasks** We fine-tune on eight commonsense and reasoning benchmarks used in Hu et al. (2023): PIQA (Bisk et al., 2020), HellaSwag (Zellers et al., 2019), WinoGrande (Sakaguchi et al., 2020), SiQA (Sap et al., 2019), OBQA (Mihaylov et al., 2018), BoolQ (Clark et al., 2019), ARC-Easy and ARC-Challenge (Clark et al., 2018). For each dataset, we train adapters for up to 30 epochs, monitor validation accuracy on the official validation set split, select the checkpoint with the highest performance, and report its test accuracy.

**Training procedure** Adapters are trained with AdamW (Loshchilov and Hutter, 2019) (learning rate $3 \times 10^{-4}$) with other default settings of `Trainer`. We use a micro-batch size of 8 per GPU (no gradient accumulation) on NVIDIA A100s, mixed precision (FP16) via `torch.cuda.amp`, weight decay $0.01$, a dropout rate of $0.1$, and the default linear scheduler. We found that introducing gradient accumulation to simulate larger effective batch sizes (e.g. accumulating 8 steps to achieve

an effective batch of 8) can shift final accuracy by several percentage points, so all reported results use no accumulation. Checkpoint save/load overhead is excluded from our reported throughput; nevertheless, since Kron-LoRA's adapters have a significantly smaller parameter footprint, its saving checkpoint time is substantially lower than that of standard LoRA by roughly 10% for each time of saving. All experiments can be performed within 24 hours, but we need to set epoch = 16 for Hellaswag.

**Continual-learning protocol**  To assess forgetting, we fine-tune adapters sequentially on two tasks and then evaluate on the first task's test set. We use the following schedules:

- **ARC-Challenge↔ARC-Easy:** 15 epochs on ARC-Challenge followed by 15 epochs on ARC-Easy, and vice versa.
- **HellaSwag↔ARC-Easy:** 10 epochs on HellaSwag followed by 15 epochs on ARC-Easy, and vice versa.

**Evaluation metrics**  We report (1) test accuracy, (2) training dynamics, (3) training throughput (examples/sec, excluding I/O), and (4) peak and intermediate GPU memory.

## 5 RESULTS

### 5.1 PARAMETER EFFICIENCY VS. ACCURACY

To better understand the performance trade-offs between different adapter strategies, we evaluated KronA and LoRA-8 on the Mistral-7B model across eight standard multiple-choice reasoning benchmarks: PIQA, HellaSwag (HS), WinoGrande (WG), ARC-Easy (ARC-E), ARC-Challenge (ARC-C), OpenBookQA (OBQA), SocialIQA (SIQA), and BoolQ. Results are summarized in Table 1.

LoRA-8 achieves a strong average accuracy of 74.39%, with competitive scores across all tasks. In contrast, KronA lags significantly behind, with an average accuracy of only 54.74% — roughly 20 percentage points lower. This trend is consistent across all benchmarks, with the gap being especially large on OBQA and SIQA, where KronA struggles to surpass 45%.

It is worth noting that KronA uses far fewer trainable parameters (2.28M) compared to LoRA-8 (21.26M), since the decomposition is constrained to factors that are the closest integers around the square root of the adapter dimension, as proposed in the original paper. This factorization yields the most parameter-efficient configuration possible under the KronA design, but at the cost of a substantial drop in task accuracy.

We also observed that similar performance patterns hold across other backbone models (DistilBERT, LLaMA-2-7B, and LLaMA-3-7B), where KronA consistently underperforms LoRA by a large margin, suggesting that the observed gap is not specific to Mistral-7B.

Table 1: Mistral-7B test accuracy (%) comparing KronA and LoRA-8. KronA performs ~20% below LoRA-8. HS = HellaSwag, WG = WinoGrande. Avg. = average over all 8 benchmarks.

| Adapter | #Params | Avg. | PIQA | HS | WG | ARC-E | ARC-C | OBQA | SIQA | BoolQ |
|---------|---------|------|------|-----|-----|-------|-------|------|------|-------|
| LoRA-8 | 21.26 M | 74.39 | 85.09 | 83.85 | 80.58 | 74.74 | 54.52 | 70.60 | 64.59 | 81.13 |
| KronA | 2.28 M | 54.74 | 72.36 | 70.96 | 67.56 | 51.40 | 34.78 | 43.60 | 45.04 | 54.22 |

Since KronA by itself does not perform well—achieving extreme parameter efficiency but at the cost of substantial accuracy loss—we next investigate whether combining Kronecker factorization with LoRA can yield a more favorable balance. This leads us to Kron-LoRA, which integrates the parameter savings of Kronecker decomposition with the representational capacity of LoRA.

We compared Kron-LoRA against LoRA across four backbone models—DistilBERT, Mistral-7B, LLaMA-2-7B, and LLaMA-3-8B—on the same eight multiple-choice reasoning benchmarks: PIQA, HellaSwag (HS), WinoGrande (WG), ARC-Easy (ARC-E), ARC-Challenge (ARC-C), Open-BookQA (OBQA), SocialIQA (SIQA), and BoolQ. Results are presented in Tables 2 and 3.

Across all four backbones, Kron-LoRA achieves performance comparable to—or in several cases better than—LoRA, while requiring only 25–30% of the parameters. On DistilBERT, Kron-LoRA (0.84M parameters) reaches 57.57% average accuracy, marginally surpassing LoRA-16 (57.06%) despite using less than half the parameters.

On Mistral-7B, Kron-LoRA (5.74M parameters) achieves 75.22% average accuracy, outperforming LoRA-8's 74.39% with only 27% of the parameters. A task-level breakdown shows notable improvements on HellaSwag (+2.30 pp) and ARC-Easy (+1.93 pp), while performance on the remaining benchmarks remains comparable.

On LLaMA-2-7B, Kron-LoRA reaches 73.78% average accuracy, slightly higher than LoRA-8 (73.62%), with gains on PIQA, HellaSwag, and ARC-Easy. Minor drops are observed on Wino-Grande and SIQA, likely due to the datasets' sensitivity to subtle context and social reasoning cues that may be harder for the compressed Kronecker factorization to capture at this rank.

Finally, on LLaMA-3-8B, Kron-LoRA achieves 73.48% average accuracy versus LoRA-8's 72.71%, with consistent gains on PIQA, HellaSwag, ARC-Easy, and BoolQ. A small decrease is observed on OBQA, which may reflect dataset-specific variance rather than a systematic limitation of Kron-LoRA.

Table 2: Test accuracy (%) on reasoning and ARC benchmarks (Part 1). HS = HellaSwag, WG = WinoGrande.

| Model | Adapter | #Params | PIQA | HS | WG | ARC-E | ARC-C |
|-------|---------|---------|------|-----|-----|-------|-------|
| DistilBERT | LoRA-8 | 1.25 M | 65.56 | 25.84 | 50.20 | 50.53 | 34.78 |
| | LoRA-16 | 1.92 M | 65.40 | 36.33 | 51.46 | 53.86 | 35.79 |
| | Kron-LoRA | 0.84 M | 65.83 | 36.09 | 52.01 | 52.46 | 39.13 |
| Mistral-7B | LoRA-8 | 21.26 M | 85.09 | 83.85 | 80.58 | 74.74 | 54.52 |
| | Kron-LoRA | 5.74 M | 85.85 | 85.32 | 80.66 | 76.67 | 54.85 |
| LLaMA-2-7B | LoRA-8 | 20.28 M | 81.83 | 86.19 | 74.03 | 80.70 | 57.85 |
| | Kron-LoRA | 5.31 M | 82.05 | 86.76 | 73.16 | 81.40 | 57.85 |
| LLaMA-3-8B | LoRA-8 | 22.03 M | 82.70 | 88.90 | 68.98 | 76.49 | 54.18 |
| | Kron-LoRA | 5.84 M | 84.00 | 89.41 | 69.92 | 77.19 | 55.52 |

Table 3: Test accuracy (%) on commonsense benchmarks (Part 2). Avg. = average over all eight benchmarks.

| Model | Adapter | OBQA | SIQA | BoolQ | Avg. |
|-------|---------|------|------|-------|------|
| DistilBERT | LoRA-8 | 52.00 | 33.78 | 99.79 | 51.56 |
| | LoRA-16 | 54.20 | 59.52 | 99.91 | 57.06 |
| | Kron-LoRA ((**Ours**) | 55.20 | 59.93 | 99.91 | **57.57** |
| Mistral-7B | LoRA-8 | 70.60 | 64.59 | 81.13 | 74.39 |
| | Kron-LoRA | 71.80 | 65.20 | 81.41 | **75.22** |
| LLaMA-2-7B | LoRA-8 | 66.80 | 61.31 | 80.28 | 73.62 |
| | Kron-LoRA | 67.40 | 61.11 | 80.52 | **73.78** |
| LLaMA-3-8B | LoRA-8 | 68.80 | 58.29 | 83.30 | 72.71 |
| | Kron-LoRA | 68.20 | 59.26 | 84.31 | **73.48** |

While most of our experiments focused on rank = 8, we also conducted preliminary trials at higher ranks. They also expected to yield similar results, as Kron-LoRA continued to perform on par with LoRA.

## 5.2 TRAINING DYNAMICS

Figure 1 shows validation accuracy on HellaSwag over 16 epochs for Kron-LoRA ($d_{A2} = 16, r = 8$) and LoRA adapters of ranks 4, 8, and 16 on Mistral-7B. Despite using only 27% of LoRA-8's parameters—and half of LoRA-4's—Kron-LoRA outperforms LoRA-8 after the first two epochs and maintains its lead for the remainder of training. Although LoRA-16 achieves the highest accuracy, it requires over six times more adapter parameters than Kron-LoRA. Furthermore, Kron-LoRA's learning curve is smoother, suggesting that the Kronecker-structured factorization provides an implicit regularization effect that stabilizes optimization and accelerates convergence.

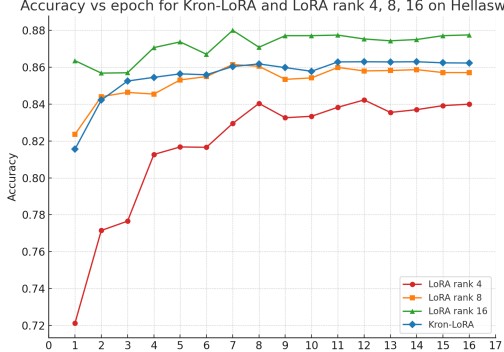

Figure 1: Validation accuracy on HellaSwag over 16 epochs for Kron-LoRA and LoRA adapters of ranks 4, 8, and 16 on Mistral-7B.

## 5.3 SPEED AND MEMORY

Table 4 compares training throughput and GPU memory usage for LoRA-8 and Kron-LoRA on PIQA with LLaMA-2-7B:

Table 4: Throughput (examples/sec) and memory (MiB) on PIQA (LLaMA-2-7B).

| Adapter | Throughput (ex/s) | Peak mem. (MiB) | Intermediate mem. (MiB) |
|---------|-------------------|-----------------|-------------------------|
| LoRA-8 | 29.28 | 28050.54 | 26594.06 |
| Kron-LoRA | 27.04 | 27822.68 | 26493.38 |

**Throughput**   Kron-LoRA processes 27.04 ex/s versus LoRA-8's 29.28 ex/s (a 7.65% slowdown). The modest overhead stems from Kron-LoRA's three matrix operations per forward pass (two `torch.matmul` and one `nn.Linear`) versus LoRA's two `nn.Linear` calls.

**Peak memory**   Kron-LoRA peaks at 27822.68 MiB, saving 227.86 MiB (0.8%) compared to LoRA-8's 28050.54 MiB due to its much smaller adapter tensors, despite the extra reshape and batched kernels.

**Intermediate memory**   Kron-LoRA requires 26493.38 MiB for activations and gradients—100.68 MiB (0.4%) less than LoRA-8's 26594.06 MiB—reflecting that, despite the additional Kronecker reshaping and batched operations, the smaller adapter tensors yield a net intermediate-memory saving.

Overall, Kron-LoRA incurs only a modest throughput overhead (5–8%) while reducing peak memory usage by approximately 0.8%, suggesting a reasonable trade-off between speed and memory efficiency for large-scale fine-tuning.

## 5.4 CONTINUAL LEARNING (FORGETTING)

To measure forgetting, we sequentially fine-tune adapters on two tasks (A→B) and then evaluate accuracy on task A's test set. Table 5 shows results for the closely related ARC-Challenge ↔ ARC-Easy sequence and the more heterogeneous ARC-Easy ↔ HellaSwag sequence. Interestingly, when tasks are highly similar, both Kron-LoRA and LoRA-8 can exhibit not only limited forgetting but even positive transfer, whereas under larger domain shifts their robustness diverges.

Table 5: Accuracy for sequential fine-tuning tasks under different sequences on LLaMA-3-8B. T1 and T2 denote the first and second tasks in the sequence, respectively. Columns show performance on the first task after training on T1 (T1→T1), on the second task after training on T2 (T2→T2), and on the first task after training on T2 (T2→T1). The fourth column shows the accuracy drop on the first task ($\Delta$T1 = T2→T1 − T1→T1).

| Sequence | Kron-LoRA (%) | | | | LoRA-8 (%) | | | |
|---|---|---|---|---|---|---|---|---|
| | T1→T1 | T2→T2 | T2→T1 | $\Delta$ T1 | T1→T1 | T2→T2 | T2→T1 | $\Delta$ T1 |
| ARC-C→ARC-E | 48.83 | 74.74 | 52.51 | +3.68 | 48.16 | 75.96 | 51.84 | +3.68 |
| ARC-E→ARC-C | 77.02 | 56.19 | 72.98 | -4.04 | 76.18 | 55.52 | 71.96 | -4.22 |
| ARC-E→HS | 73.51 | 87.13 | 58.77 | -14.74 | 72.28 | 87.32 | 3.68 | -68.60 |
| HS→ARC-E | 87.23 | 72.81 | 83.08 | -4.15 | 87.00 | 71.40 | 82.63 | -4.37 |

**Robust retention under task similarity** In the ARC-Challenge→ARC-Easy ordering, both Kron-LoRA and LoRA-8 slightly *improve* performance on the first task after training on the second (+3.68 pp), indicating beneficial transfer between the two datasets. In the reverse ARC-Easy→ARC-Challenge sequence, both methods experience a modest drop, with Kron-LoRA losing slightly less (–4.04 pp) than LoRA-8 (–4.22 pp). These results suggest that, under high task similarity, both approaches preserve and leverage shared representations effectively, with Kron-LoRA providing a small but consistent advantage in retention.

**Increased interference under domain shift** For the heterogeneous ARC-Easy→HellaSwag sequence, Kron-LoRA retains 58.77% (–14.74 pp), while LoRA-8 collapses almost entirely (–68.60 pp). In the reverse HellaSwag→ARC-Easy direction, both methods suffer modest forgetting (–4.15 pp vs. –4.37 pp). These results highlight that Kron-LoRA is substantially more robust under large domain shifts, mitigating catastrophic interference that severely affects LoRA-8.

**Regularization and optimization effects** We found that Kron-LoRA benefits noticeably from weight decay, which acts as a regularizer and helps stabilize retention across both similar and heterogeneous tasks. Moreover, Kron-LoRA's performance indicates that its structured adaptation provides a better inductive bias for cross-domain generalization, though additional strategies such as task-specific adapter reinitialization or merging could further reduce forgetting.

## 6 BROADER APPLICABILITY AND FUTURE WORK

Below, we outline several cross-disciplinary applications of Kron-LoRA that demonstrate its broader impact beyond NLP:

1. **Edge-scale medical imaging.** Radiology models must often be fine-tuned per hospital or modality (MRI, CT, ultrasound), yet storing dozens of adapters on a PACS server or embedded probe is impractical. Kron-LoRA's $4\times$ smaller factors allow shipping and switching task- or device-specific adapters in seconds—even on low-power ARM or FPGA hardware.

2. **On-device mobile models.** Smartphones and wearables increasingly host large language or vision models for tasks like translation, summarization, or AR guidance. Yet frequent task-specific updates (per app, per user) are infeasible when adapters span tens of MB. Kron-LoRA's compact Kronecker factors ($\sim$1–2 MB) can be shipped as lightweight app updates, cached per-user, or swapped across applications with negligible storage overhead.

3. **Multi-physics surrogate modeling.** High-fidelity simulators in climate science, fluid dynamics, or materials design frequently require fine-tuning to new boundary conditions, but storing a full model per scenario is prohibitively expensive. With Kron-LoRA, a single base network can host dozens of tiny adapters—one per regime—so that thousands can reside in GPU memory for real-time interpolation across parameters.

4. **Robotics & control.** A robot arm may need distinct control policies for assembly, painting, and inspection, yet continual fine-tuning risks catastrophic forgetting and swapping large networks adds latency. By attaching separate Kron-LoRA adapters (∼1 MB each) per skill, the robot can load any policy in milliseconds.

5. **Neuromorphic & photonic accelerators.** Emerging hardware (e.g. spiking neural nets, photonic matrix multiplies) favors low-rank, structured updates to minimize on-chip memory and routing complexity. Kron-LoRA's factorization maps naturally to 2D crossbar arrays (for the Kronecker factor) and digital peripheral logic (for the low-rank factors), squeezing adapters into tiny on-chip buffers.

6. **Federated & privacy-preserving learning.** In federated settings (e.g. personal keyboard models, health data), uploading full LoRA adapters (tens of MB) strains bandwidth and raises privacy concerns. Kron-LoRA requires only a few-MB factors per client—minimizing communication and simplifying secure server-side aggregation.

These vignettes show that Kron-LoRA is more than an NLP trick; it is a broadly applicable, structured, multi-task adapter framework poised to transform fine-tuning across disciplines.

## 7 CONCLUSION

We have introduced *Kron-LoRA*, a simple drop-in adapter that combines Kronecker-structured updates with low-rank LoRA compression to achieve up to $4\times$ fewer parameters than standard LoRA while preserving similar accuracy. To the best of our knowledge, Kron-LoRA is the first method that explicitly integrates Kronecker factorization with LoRA—an original combination not previously explored in parameter-efficient fine-tuning or in the broader matrix approximation literature. Our empirical results demonstrate that Kron-LoRA matches or exceeds LoRA baselines in multi-task accuracy, incurs only a 5-8% training-speed overhead, reduces memory usage, and maintains competitive or slightly improved performance on similar tasks, while experiencing much smaller drops than LoRA-8 on heterogeneous task sequences.

**Key takeaway:** As the first integration of Kronecker structure with LoRA, Kron-LoRA offers a plug-and-play, scalable, and sustainable adaptation layer for large pre-trained transformers—empowering parameter-efficient and continual-learning-capable fine-tuning for similar datasets.

## ETHICS STATEMENT

**Potential benefits** Kron-LoRA's extreme parameter efficiency makes it feasible to *deploy* fine-tuned adapters on resource-constrained hardware (e.g. mobile devices, edge nodes, or IoT sensors). This can democratize access to state-of-the-art language capabilities in low-resource settings—such as small clinics, field research stations, or educational programs without dedicated GPU infrastructure—and reduce the carbon footprint of continual updates by minimizing computation and memory requirements. Moreover, in privacy-sensitive applications (e.g. personalized keyboard suggestions, on-device medical note classification, or federated learning), Kron-LoRA's tiny adapters can be exchanged instead of full model weights, lowering communication overhead and limiting the surface area for data leakage.

**Responsible usage considerations** As with all adapter-based methods, Kron-LoRA inherits biases and limitations from its underlying pre-trained model, so practitioners should evaluate downstream fairness and robustness across demographic groups and application domains. Deploying compact adapters on edge or federated platforms also raises security concerns—malicious actors could craft poisoned updates or invert small adapters to extract private signals—so secure update channels, differential privacy, and audit logs are recommended. Finally, while parameter-lean

adapters lower the barrier to customization, developers must still adhere to ethical standards, ensure transparency about model capabilities and limitations, and obtain informed consent when processing sensitive data.

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
