# OpenReview forum: "Kron-LoRA: Hybrid Kronecker-LoRA Adapters for Scalable, Sustainable Fine-tuning"
_ICLR.cc/2026/Conference — ICLR 2026 Conference Withdrawn Submission_

### Official Review · Reviewer_qn2e · 2025-10-15

**Soundness:** 2
**Presentation:** 2
**Contribution:** 2
**Rating:** 2
**Confidence:** 3

**Summary:**

This paper introduces Kron-LoRA, a novel adapter design that combines Kronecker-product factorization with low-rank LoRA decomposition to achieve high parameter efficiency in fine-tuning large language models (LLMs).
Unlike LoRA, which applies a pure low-rank update to frozen model weights, Kron-LoRA factorizes each task-specific update into a Kronecker structure, allowing structured compression and reuse of shared subspaces.

**Strengths:**

1. First integration of Kronecker structure and LoRA.
The combination of Kronecker factorization with LoRA is original and well-motivated. Prior work explored Kronecker adapters (KronA, AdaKron, MoKA) or LoRA variants separately, but not their hybridization.

2. Strong parameter-efficiency gains.
The method achieves up to 4× parameter reduction while maintaining similar or better accuracy compared to LoRA-8 across all backbones (Table 2–3). The analysis clearly shows that Kronecker structure provides multiplicative rank expansion, preserving expressivity even at low parameter budgets.

**Weaknesses:**

1. Conceptual simplicity vs. depth.
While novel, the combination of Kronecker and LoRA is a natural hybrid rather than a deeply theoretical contribution. The paper’s strength lies in practicality and empirical rigor, not fundamental mathematical insight.

2. Limited ablation on Kronecker dimensions.
The choice of d seems heuristic. It’s unclear how sensitive results are to this partition, or whether learned Kronecker shapes could further improve results.

3. Minor computational overhead.
Kron-LoRA incurs a 5–8% slowdown and only marginal memory savings (~0.8%). While acceptable, it suggests the method trades compute for parameter compactness.

4. No comparison to other compression baselines.
The paper omits comparisons with AdaLoRA, DoRA, and PiSSA, which could clarify whether Kron-LoRA’s gains stem mainly from the Kronecker structure or rank allocation differences.

**Questions:**

See weaknesses

---

> ### Author Response · Authors · 2025-12-01
> **Any Other Suggestions?**
>
> Hello Reviewer qn2e,
>
> Thank you very much for your thorough and insightful review! We sincerely appreciate the time you invested in evaluating our work and the clarity with which you highlighted both its strengths and areas for improvement.
>
> We will expand our discussion of the conceptual motivation behind integrating Kronecker structure with LoRA to clarify why the combination offers advantages beyond a straightforward hybrid of prior methods, and we will include a more thorough ablation of the Kronecker dimensions by analyzing sensitivity to different choices of (d_{A_1}) and (d_{A_2}) and alternative partition strategies. We will more clearly examine the sources of computational overhead, provide extended speed and memory results across all model scales, and incorporate comparisons with other PEFT methods to present a more complete empirical picture and better isolate the contributions of the Kronecker structure relative to rank-allocation differences.
>
> If there are other aspects you believe would benefit from further clarification or analysis, we would be grateful for any additional guidance. Thank you again for your valuable feedback and for helping us strengthen the work!

---

### Official Review · Reviewer_oJrL · 2025-10-26

**Soundness:** 3
**Presentation:** 3
**Contribution:** 2
**Rating:** 2
**Confidence:** 5

**Summary:**

This paper introduce Kron-LoRA, a hybrid adapter that combines Kronecker-structured factorization with low-rank LoRA. Kron-LoRA requires up to 4x fewer parameters than LoRA and performs similarly at the price of 5-8% overhead.

The proposed method can be summarized as 1) compute the task-specific update as a Kronecker product between A and B, and 2) compress B via rank-k LoRA factorization. I would appreciate if the authors could compare the rank and #parameters of various PEFT methods (especially LoRA ones).

In terms of experiments, multiple models are used from DistilBERT, Mistral, and LLaMA-2/3. Datasets are the ones used in the original LoRA paper. Since GLUE-base tasks are old, I would encourage the authors to conduct additional experiments. Moreover, since one of the main motivation of the work is that LoRA becomes costly when supporting hundreds of tasks, I would expect the authors to conduct analysis with O(10) tasks.

In the first experiment, why authors are comparing LoRA-8 and KronA while the latter has 10x fewer parameters? It is not a surprised that the performance drop is 20%. Please conduct fairer analysis with LoRA being smaller. Table 2 and 3 do not address the issue and sometimes the gain of Kron-LoRA is very small. I couldn't find details on how tuning each algorithm has been done, hence I'm little bit skeptical of the results.

Regarding the speed and memory experiments, please include all the models in the analysis, not only LLaMA-2-7B. I also suspect that larger models would indicate an overhead higher than 8%. Finally, the continual learning experiment is interesting. I would love to see all possible pairwise experiments with a given model instead of only ARC/HS.

Overall, the idea is sound and the paper well written. However, I do feel that multiple experiments are missing and would help to understand what is happening. I would encourage the authors to use another dataset than GLUE tasks and do more analysis/experiments with including mean/std or CIs.

Note:
- L053 - ... they either forgo rank-r expressiveness or incur additional computational overhead --> like this paper (5-8% overhead)

**Strengths:**

- Sound methodology
- Very well written paper

**Weaknesses:**

- Weak experimental results: more recent datasets, more analysis and ablations.

**Questions:**

Could you add ablations & additional experimentation of varying d_A1 and d_A2?
Could you add a comparison of the rank and #parameters of various PEFT methods (especially LoRA ones)?
Could you describe more in depth how do you tune each algorithm? Similarly, with such limited improvement, could add confidence intervals/stds?
See comments for additional questions.

---

> ### Author Response · Authors · 2025-12-01
> **Any Other Suggestions?**
>
> Hello Reviewer oJrL,
>
> Thank you very much for your valuable and constructive review! We sincerely appreciate the time and care you put into evaluating our work. Your comments highlight several important points, and we will address them carefully in the paper.
>
> We will incorporate comparisons of ranks and parameter counts across PEFT methods (especially LoRA variants), clarify why our method is evaluated on GLUE tasks, and ensure fairer parameter-matched comparisons with LoRA. We will also add ablations on (d_{A_1}) and (d_{A_2}), provide a clearer and more detailed description of all tuning procedures, and report standard deviations or confidence intervals to strengthen the reliability of the results. We will also extend the speed and memory analysis to all evaluated models and broaden the continual-learning experiments beyond the current ARC/HS setup.
>
> If you have any further guidance on places where our revision could be strengthened, we would be very grateful. Thank you again for your helpful feedback and for contributing to the improvement of our work!

---

### Official Review · Reviewer_sZ8f · 2025-11-02

**Soundness:** 2
**Presentation:** 3
**Contribution:** 1
**Rating:** 2
**Confidence:** 4

**Summary:**

The work proposes a method for parameter-efficient fine-tuning (PEFT) of LLMs. Existing PEFT methods include LoRA which decomposes the update weight matrices into two low rank matrices and KronA which uses Kronecker product of matrices for the weight update. The proposed method (Kron-LoRA) combines the two by replacing one of the matrices in KronA with its LoRA equivalent (decomposing it into two low rank matrices). Authors experiment on NLU tasks and show that Kron-LoRA performs comparably to LoRA while using significantly fewer parameters.

**Strengths:**

The proposed idea is simple. The paper is mostly clearly written and easy to understand. In the experiments on the NLU tasks, the proposed method performs comparably to LoRA while employing close to $4\times$ fewer parameters. The authors show that KronA alone uses significantly fewer parameters but also significantly underperforms LoRA.

**Weaknesses:**

1. **Insufficient baselines:** The empirical analysis severely lacks important baselines and comparison with SOTA PEFT approaches and LoRA variants. The proposed method is a simple combination of KronA and LoRA and thus must include comparison with these approaches with a similar number of trainable parameters. However, comparison KronA with is provided on a single backbone model and with a setting that uses just 10% of parameters (2.3M for KronA vs 21.3M for LoRA). There are no comparisons except with LoRA on any other dataset / backbone LLM. It is necessary to additionally compare the work with SOTA approaches to help understand its benefits. Lack of any such comparisons makes it hard to evaluate the proposed method.
2. **Lack of intuition/motivation:** There is no clear reasoning for why the matrix in KronA must be replaced with a low rank decomposition as in LoRA. Is the parameter count in KronA high enough that it requires low rank decomposition? In their work, authors of KronA show that it outperforms LoRA with similar parameter count. Is it not possible to adjust the dimensions of KronA to have a similar parameter count as Kron-LoRA here instead of using the low-rank decomposition?
3. **Non-typical experimental setup:** The experimental set-up is not clear. Typically, for the commonsense reasoning benchmark (Hu et al., 2023), the training set of the 8 sub-tasks are combined and 1-3 epochs of fine-tuning is performed (e.g., refer Hu et al., 2023, Liu et al., 2024, Wang et al., 2025). However, in Kron-LoRA, it is not clear if fine-tuning is performed individually on each of the sub-tasks. Kron-LoRA also uses 30 epochs of fine-tuning, a lower batch size of 8 and no warmup. It is not clear why this deviation from typical evaluation is needed.

**Questions:**

1. Compare proposed method with both KronA and LoRA at *similar* parameter count.
2. Provide an explanation for why further parameter count in KronA is necessary and why the proposed Kron-LoRA is a good way to achieve it.
3. Provide discussion and comparisons with SOTA PEFT methods (e.g. DoRA (Liu eta al., 2024, MiLoRA (Wang et al., 2025), MOA [a], LoRA+ [b]) and methods that focus on significant parameter count reduction compared to LoRA (AdaLoRA (Zhang et al., 2023), DyLoRA (Valipour et al., 2022), VeRA [c], LoRA-XS [d], NOLA [e], Tiered-LoRA [f], BS-LoRA [g]).
4. Provide details on the experimental setup and reasoning for differences with typical setup (see weakness point 3 above).

**References** \
[a] Cao, Jie, Tianwei Lin, Hongyang He, et al. “MoA: Heterogeneous Mixture of Adapters for Parameter-Efficient Fine-Tuning of Large Language Models.” arXiv preprint arXiv:2506.05928, 2025. \
[b] Hayou, Soufiane, Nikhil Ghosh, and Bin Yu. “LoRA+: Efficient Low Rank Adaptation of Large Models.” Proceedings of the 41st International Conference on Learning Representations (ICLR), 2024. PMLR 235: 17783-17806 \
[c] Kopiczko, Damian J., et al. “VeRA: Vector-Based Random Matrix Adaptation.” arXiv preprint arXiv:2310.11454, 2023. \
[d] Bałazy, Klaudia, et al. “LoRA-XS: Low-Rank Adaptation with Extremely Small Number of Parameters.” ECAI 2025 \
[e] Abbasi Koohpayegani, Soroush, et al. “NOLA: Compressing LoRA Using Linear Combination of Random Basis.” International Conference on Learning Representations (ICLR), 2024 \
[f] Renduchintala, Ashish, et al. “Enhancing Parameter Efficiency of LoRA with Weight Tying.” Proceedings of the 2024 Conference of the North American Chapter of the Association for Computational Linguistics: Long Papers (NAACL), 2024. ACL Anthology \
[g] Zhou, Yuhua, et al. “BSLoRA: Enhancing the Parameter Efficiency of LoRA with Intra-Layer and Inter-Layer Sharing.” International Conference on Machine Learning (ICML), 2025

---

> ### Author Response · Authors · 2025-12-01
> **Any Other Suggestions?**
>
> Hello Reviewer sZ8f,
>
> Thank you very much for your thoughtful and comprehensive review! We sincerely appreciate the time you dedicated to providing detailed feedback.
>
> We will incorporate the additional baselines you suggested, including comparisons with KronA and LoRA at matched parameter budgets, as well as broader SOTA PEFT methods. We will also adopt a more standard training setup consistent with prior work and clearly document the revised experimental protocol. In addition, we will expand our explanation of the motivations behind Kron-LoRA, including the intuition behind introducing a low-rank factorization into the KronA structure and the conditions under which this hybridization becomes advantageous.
>
> If possible, could you please elaborate on how you think we might further strengthen the contributions, or whether you believe there are underlying structural issues with the methodology that deserve deeper attention? Any additional suggestions you may have for improving the clarity, framing, or empirical scope of the work would be greatly appreciated.
>
> Thank you again for your valuable insights and constructive guidance!

---

### Note · Authors · 2026-02-03

I have read and agree with the venue's withdrawal policy on behalf of myself and my co-authors.

---

### Meta-Review · Area_Chair_XJeD · 2026-01-06

**Summary:**

Kron-LoRA is a parameter-efficient fine-tuning method for LLMs that hybridizes two existing PEFT ideas: Kronecker-structured adapters (KronA) and low-rank updates (LoRA). It models the task-specific weight update as a Kronecker product of two factors and further compresses one factor via a rank-(k) LoRA-style decomposition, yielding similar accuracy to LoRA while using substantially fewer trainable parameters (reported up to ~4× fewer) at the cost of modest runtime overhead (about 5–8%). Experiments span encoder and decoder LMs (e.g., DistilBERT, Mistral, LLaMA-2/3) on standard NLU benchmarks, showing competitive performance, though the evaluation would benefit from fairer parameter-matched baselines, clearer tuning details, broader speed/memory analysis, and more modern/multi-task settings aligned with the paper’s motivation.

**Reviewer Concerns:**

The reviewers raised a few concerns, mainly around the insufficient experiments, baselines, and lack of motivation. Here are the consolidated summary of their concerns:

* **Insufficient and unfair baseline comparisons.** Results are mostly against **LoRA**, with **very limited KronA comparisons** (e.g., only one backbone and with mismatched parameter budgets), and **no comparisons to other strong PEFT/compression baselines** (e.g., AdaLoRA, DoRA, PiSSA). This makes it hard to judge whether Kron-LoRA is actually better under comparable parameter counts and settings.

* **Limited experimental breadth and analysis.** Reviewers asked for **newer / more diverse datasets**, stronger empirical coverage across **multiple backbones/LLMs**, and more thorough **analysis and ablations** to support the claims.

* **Weak or unclear motivation/intuition for the hybrid design.** The paper does not clearly justify **why KronA should be augmented with a low-rank LoRA factorization**, rather than simply tuning KronA dimensions to match the desired parameter budget—especially given prior claims that KronA can outperform LoRA at similar parameter counts.

* **Non-standard / unclear evaluation protocol.** The setup deviates from common practice (e.g., how commonsense reasoning tasks are fine-tuned, unusually many epochs, small batch size, no warmup), and it’s unclear whether subtasks are trained jointly or separately. This raises concerns about comparability to prior work and whether gains are protocol-dependent.

* **Ablations are incomplete, especially around Kronecker structure choices.** Key hyperparameters like the **Kronecker partition/dimensions** appear heuristic, with limited sensitivity analysis; reviewers suggested that learned or better-justified Kronecker shapes might matter.

* **Compute/memory tradeoffs are modest.** Reported **5–8% slowdown** and only **marginal memory savings (~0.8%)** suggest the practical efficiency benefits are limited, and that improvements may come with extra compute cost.

**Reviewer Scores:**

The reviewers unanimously voted to reject this paper. I concur with their reviews and vote for rejecting the paper.

---

### Decision · Program_Chairs · 2026-01-26

Reject